# Differences in Improvement of Physical Function in Older Adults with Long-Term Care Insurance with and without Falls: A Retrospective Cohort Study

**DOI:** 10.3390/healthcare11182558

**Published:** 2023-09-15

**Authors:** Masahiro Kitamura, Junichi Umeo, Kyohei Kurihara, Takuji Yamato, Takayuki Nagasaki, Katsuhiko Mizota, Haruki Kogo, Shinichi Tanaka, Takashi Yoshizawa

**Affiliations:** 1School of Physical Therapy, Faculty of Rehabilitation, Reiwa Health Sciences University, 2-1-12 Wajirooka, Higashi-ku, Fukuoka 811-0213, Japan; 2Kizuna Daycare Center, 1399-1 Imai, Yukuhashi 824-0018, Japan; junichi-umeo@reha-kizuna.com (J.U.);

**Keywords:** physical function, fall, older adults, long-term care insurance

## Abstract

(1) Background: This study examined the differences in changes in physical function with and without falls after daycare use among frail older adults with long-term care insurance (LTCI). (2) Methods: In this retrospective cohort study, 82 of 96 consecutive daycare center users met the inclusion criteria. The participants were divided into two groups based on the presence or absence of falls 6–12 months after use. Participant characteristics in the fall and non-fall groups and physical function at baseline and six months in each group were compared. Using analysis of covariance, we analyzed physical function and its changes between the two groups, and cut-off values were calculated using receiver operating characteristic curves. (3) Results: Gait speed, timed up-and-go test, and 30 s chair stand test (CS30) improved significantly over six months in the no-fall group (*n* = 70) and all participants (*n* = 82) (*p* < 0.01). Gait speed in the fall group (*n* = 12) improved significantly over six months (*p* = 0.04). The fall group had significantly lower adjusted ΔCS30 scores than the no-fall group (*p* = 0.03), with a cutoff value of 2 (*p* = 0.024). (4) Conclusions: In older adults with LTCI, physical function with and without falls after daycare use differed by ΔCS30, with a cutoff value of 2.

## 1. Introduction

Among falls and fractures in older adults worldwide, at least one fall per year occurs in 30% of those over 65 years of age [1], and 10% of these falls result in fractures that cause difficulties in the activities of daily living (ADL) [2,3]. The annual fall rate among community-dwelling older adults in Japan is approximately 20% [4], and fractures due to falls often require long-term care insurance (LTCI) because of ADL difficulties [5]. Moreover, an increase in healthcare and long-term care expenditures associated with hospitalization and the use of LTCI pose serious problems [6]. Furthermore, older adults with LTCI have a fall rate of 25% and are more likely to fall than those without [7]. Repeated falls cause a significant decline in ADL and serious illness due to refracture and fear; therefore, measures to reduce future falls are extremely important [8,9,10]. Among fall risk factors, physical function deterioration, such as deterioration in lower extremity muscle strength, walking ability, and balance, has been reported [2,11].

Exercise has been shown to be effective in improving physical function in older adults, including muscle strength, balance, and walking ability [12]. Improvements in physical function have also been reported in older adults with frailty after a six-month exercise program that included resistance, balance, and gait training [13,14,15,16,17,18]. Furthermore, the effectiveness of lower-extremity strength, gait, and balance training in preventing falls in older adults has been reported [12,19,20,21,22].

However, the rate of falls among older adults with LTCI after using daycare is problematic, as high as 18.3% over a 3-month period [23]. In addition, there is a lack of research on the relationship between improvement in their physical function and falls. Furthermore, although the presence or absence of falls after an exercise program is important, very few studies have investigated the difference in change in physical function between the presence and absence of falls. We hypothesized that the fall group of older adults with LTCI would show less improvement in physical function with an exercise program in daycares than the no-fall group would. If we can identify those indicators of physical function that are less likely to change, we can revise the exercise program to improve those physical functions, and possibly prevent falls after daycare use. Therefore, this study aimed to retrospectively examine the differences in physical function and its changes in LTCI older adults with and without falls after using daycare.

## 2. Materials and Methods

### 2.1. Design and Participants

This was a retrospective cohort study. The participants were 96 consecutive new users of the daycare center between April 2018 and December 2020. We included participants who could walk with or without aid, were aged 65 years or older, had used LTCI, and had been using the center for 12 months. We excluded those with dementia who scored less than 24 on the mini-mental state examination [24]. Age, sex, weight, LTCI level [25], comorbidities, history of falls [26], and physical functions such as gait speed [27], timed up-and-go test (TUG) [28], and 30 s chair stand test (CS30) were also investigated [29]. This study complied with the Declaration of Helsinki and was approved by the Reiwa Health Sciences University Ethics Committee (approval no. 22-009) and obtained the informed consent of each participant.

### 2.2. LTCI Levels

The LTCI system in Japan was introduced to meet the demands of older adults, based on the social insurance system [25]. The LTCI level is determined by the LTCI committee of the community in which the participants live and is available to persons aged 65 years and older and 40–64 years with certain diseases. Support level 1 is for people who are independent in ADL but need supervision in activities such as shopping. Support level 2 is for people whose ability to walk is impaired because of lower-limb muscle weakness. Care level 1 refers to individuals requiring simple care for ADL. Care level 2 refers to people who need care for ADL such as eating, urinating, and bathing. Care level 3 refers to people who use walking aids or wheelchairs for mobility and require considerable care for ADL. Care level 4 refers to those who use a wheelchair for mobility and cannot perform ADL without care. Care Level 5 refers to patients who are bedridden, unable to communicate, or unable to eat alone. The LTCI level was investigated by two physical therapists based on clinical data from the participants.

### 2.3. History of Falls

An older adult’s fall was determined by “unintentionally coming to the ground or at a lower level for reasons other than as a consequence of sustaining a violent blow, loss of consciousness, sudden onset of paralysis as in stroke, or an epileptic seizure” [30]. The presence of falls in daily life was investigated by two physical therapists, one nurse, and one caregiver during each weekly daycare center visit. If a fall occurred, a questionnaire was administered to investigate the date and time of the fall and its causes. 

### 2.4. Physical Function

The main outcomes of the study were changes in physical function between baseline and 6 months later: Δgait speed, ΔTUG, and ΔCS30. Gait speed, a measure of walking ability, was measured by a physical therapist using a stopwatch and defined as the time required to walk 5 m at a normal speed [27]. A 1 m acceleration and deceleration walkway was set before and after the 5 m walkway. The TUG test was used to assess balance and mobility [28]. In this test, the measurer asked the participant to “get up from the chair, walk 3 m back and forth at a normal pace, and then sit back down on the chair,” and measured the time with a stopwatch. The CS30 was used as a measure of lower extremity muscle strength [29]. The measurer prepared a chair with a seat height of 40 cm, asked the participants to “stand and sit on a chair for 30 s as many times as they could without using their arms,” and measured the number of times they could stand. These physical function measurements were performed by two physical therapists based on the measurement manual.

### 2.5. Exercise Program

The exercise program at the daycare center consisted of multicomponent exercises: (1) warm-ups such as stretching and hot packs; (2) lower extremity resistance training using body weight and machines; (3) aerobic exercise using an ergometer; (4) balance training using a ball; and (5) repetitive standing exercises [30]. These exercises were performed in groups. Initial and periodic evaluations, exercise program planning, and exercise prescriptions were set by the physical therapist with confirmation from a family doctor. Regarding exercise intensity, for aerobic exercise, the rating of perceived exertion (RPE) was “fairly light to somewhat hard” [31], and for resistance training, “approximately 60% intensity of maximal load × 10–15 times × three sets” [32]. Regarding time and frequency, the total time was 120 min, with 20 min for each practice session plus rest, and the frequency was 1–2 times per week, with the assistance of two physical therapists, one nurse, and one caregiver. The exercise program was implemented for a period of at least 6 months. Also, the purpose of the program and participation rules were explained at the beginning of use, and the staff recorded attendance for each use.

### 2.6. Sample Size

This study’s sample size used G-Power 3.1 software. The total sample size was at least 36, calculated with an effect size of 0.96, α error of 0.05, and power of 0.8 [33].

### 2.7. Statistical Analysis

Participant characteristics and clinical parameter values were reported as mean ± standard deviation for continuous variables and as percentages for categorical variables. Statistical analysis was performed after assessing the normal distribution of the data using the Shapiro–Wilk test. Categorical variables are reported as numbers (%) and variables as mean (±standard deviation) in the participant characteristics and evaluated parameter values. The participants were divided into two groups based on the presence or absence of falls 6–12 months after the start of the daycare center. Unpaired *t*-tests, Mann–Whitney U-tests, and chi-square tests were used to compare participant characteristics and clinical parameters between the fall and no-fall groups. Paired *t*-tests and Wilcoxon tests were used to compare physical function at baseline and six months in the fall group, no-fall group, and all participants. Analysis of covariance was used to compare the differences in physical function and changes between the two groups. The covariates used were age, sex, and LTCI level between the two groups. A receiver operating characteristic (ROC) curve was used for fall identification and the area under the curve (AUC) was calculated. The Youden index determines the cut-off value for physical function in the presence of falls. AUC values > 0.9 indicate high precision, 0.7–0.9 indicate medium precision, and <0.7 indicate low precision [34]. Statistical significance was set at *p* value < 0.05. Statistical analyses were performed using the IBM SPSS 25.0 J statistical software (IBM SPSS Japan, Inc., Tokyo, Japan).

## 3. Results

### 3.1. Participant Flow

The flowchart of the participants included in this study is shown in Figure 1. Of the 96 participants, 83 met the inclusion criteria for this study; of the 83 participants, those with dementia were excluded, resulting in a final number of 82 participants (fall group, *n* = 12; no-fall group, *n* = 70; fall rate, 14.6%). The six-month exercise program participation rate for the participants analyzed was 83%.

### 3.2. Characteristics of Fall Group and No-Fall Group

Table 1 presents the characteristics such as age, sex, weight, LTCI, and comorbidity of the fall and no-fall groups. There were no significant differences between the two groups.

### 3.3. Pre- and Post-Comparison of Physical Function for Each Group 

Results are presented below for a before and after comparison of each group’s physical function. Comparing physical function at the start and six months later, there were significant improvements in gait speed (0.93 ± 0.33 m/s vs. 1.14 ± 0.34 m/s; *p* < 0.001, 0.94 ± 0.32 m/s vs. 1.17 ± 0.36 m/s; *p* < 0.001), TUG (13.3 ± 8.07 s vs. 11.4 ± 6.1 s; *p* = 0.001, 13.1 ± 7.8 s vs. 11.1 ± 5.8 s; *p* < 0.001), and CS30 (12.8 ± 7.4 times vs. 15.3 ± 6.1 times; *p* < 0.001, 13.3 ± 7.5 times vs. 15.3 ± 6.1 times; *p* = 0.001) in the no-fall group and all participants, respectively. The fall group showed a significant improvement in gait speed six months after starting at the daycare center (0.99 ± 0.32 m/s vs. 1.34 ± 0.41 m/s; *p* = 0.04), with no significant improvement in TUG and CS30 (10.9 ± 4.8 s vs. 9.6 ± 3.7 s; *p* = 0.06, 17.2 ± 6.6 times vs. 15.7 ± 6.2 times; *p* = 0.29). 

### 3.4. Two-Group Comparison of Physical Function: Start, 6 Months, and Changes during That Period 

A comparison of the physical function and the main outcome, Δphysical function, between the fall and non-fall groups is presented in Table 2. The comparison reflected that the fall group had a significantly lower ΔCS30 value than the no-fall group (*p* = 0.02). After adjusting for age, sex, and LTCI, ΔCS30 was significantly different between the two groups (*p* = 0.03).

### 3.5. Cut-Off Value for the Presence of Falls

The cut-off value of ΔCS30 for the presence of a fall is shown in Figure 2. The cut-off value was calculated two times (AUC, 0.704; *p* = 0.024).

## 4. Discussion

To the best of our knowledge, this is the first report to show differences in changes in physical function depending on the presence or absence of falls in older Japanese adults with LTCI after using daycare. The results showed that the rate of falls among older adults with LTCI after 6–12 months of daycare center use was 14.6% and that the fall group had a lower ΔCS30, the change in lower limb muscle strength, than the no-fall group. In addition, the cut-off value for the presence of falls was two times.

### 4.1. Fall Rate and Improvement in Physical Function 

The fall rate of 14.6% per six months in this study was lower than the 44.5% per year for home-discharged adults after stroke [35] but higher than the 10–20% per year for community-dwelling older adults [4,36]. The fall rate in older adults with LTCI was 25.3% per year and 18.3% per 3 months [7,23], similar to the results of our study.

Regarding the significant improvement in gait speed, TUG, and CS30 for the no-fall group and all participants, and gait speed for the fall group, physical function improvement in older adults with LTCI has been reported with the use of a daycare center for six months [16,17,18]. Thus, the exercise program at the daycare center is expected to improve the physical function of most participants. However, regarding the failure to show improvement in TUG and CS30 tests in the fall group, older adults at risk of falling reported higher TUG and lower CS30 values than those at no risk [33].

### 4.2. Cut-Off Value for the Presence of Fall

To the best of our knowledge, there have been no reports on the differences in improvements in physical function between LTCI older adults with and without falls after using daycare. In this study, Δgait speed and ΔTUG showed no difference in change in physical function between the fall and no-fall groups. In a reported randomized controlled trial (RCT) of older adults with LTCI, an exercise program increased walking ability and balance [37], and it is possible that the present study had similar effects on gait speed and TUG in both groups. On the other hand, the ΔCS30 was shown as the difference in physical function in the presence of a fall, and its cutoff value was two times, which was a novelty of this study. CS30 is a screening test for fall risk in older adults [38,39]. The fall risk group of pre-frail older adults reported a lower CS30 than the no-fall risk group [33]. In community-dwelling older adults, the known CS30 cut-off for fall risk is 14.5 times [40]. Furthermore, the reference value for CS30 in the 80s was 15 times [41]. The CS30 at six months in the no-fall group was 15.3 times by exercise, exceeding the cut-off value of a previous study and reaching the reference value [40,41]. Also, the minimal clinically important difference (MCID) for CS30 in older adults with orthopedic disease was two times, similar to the cutoff value of two times in this study [42]. ΔCS30 may predict the risk of falling, which was not captured by the cut-off value of CS30. Therefore, in older adults with LTCI, failure to improve ΔCS30 more than twice with exercise indicated the possibility of future falls. In clinical practice, an increase in CS30 of two or more times from the start through an exercise program may be important in preventing falls.

### 4.3. Limitations

This study had some limitations. This study was conducted in a daycare center with a small sample size. In addition, we were unable to investigate confounding factors related to fall risk, such as educational history, medications, nutrition, smoking, and environmental modifications. Thus, we were unable to consider the impact of these factors on fall risk [43]. Moreover, because participants were not investigated for their use of LTCI services other than those at the center, we were unable to examine the impact of their use of these services on falls and physical function. The compliance rate for the exercise program has not been investigated and there are some missing data, and interpretation of this study should take this into account.

## 5. Conclusions

The fall rate for older adults with LTCI during 6–12 months of daycare center use was 14.6%. The difference in physical function with and without falls after using daycare showed a ΔCS30, with a cut-off value of two times. Failure to improve ΔCS30 more than twice after a daycare-based exercise program may result in a future fall.

## Figures and Tables

**Figure 1 healthcare-11-02558-f001:**
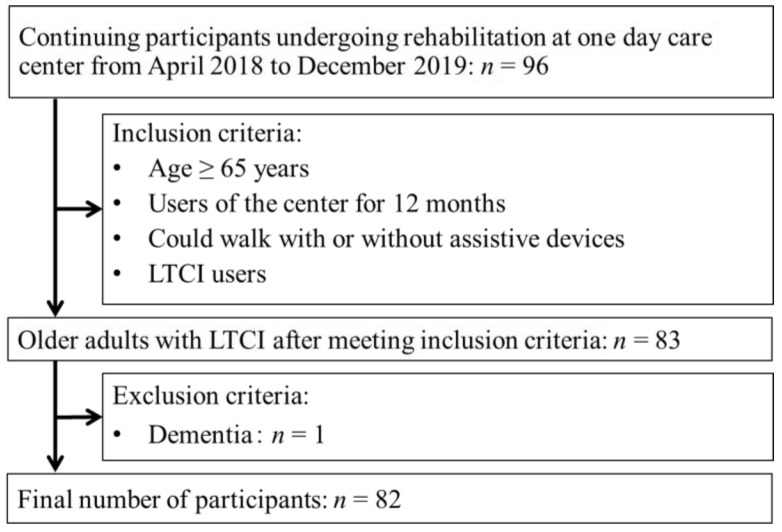
Participant flow.

**Figure 2 healthcare-11-02558-f002:**
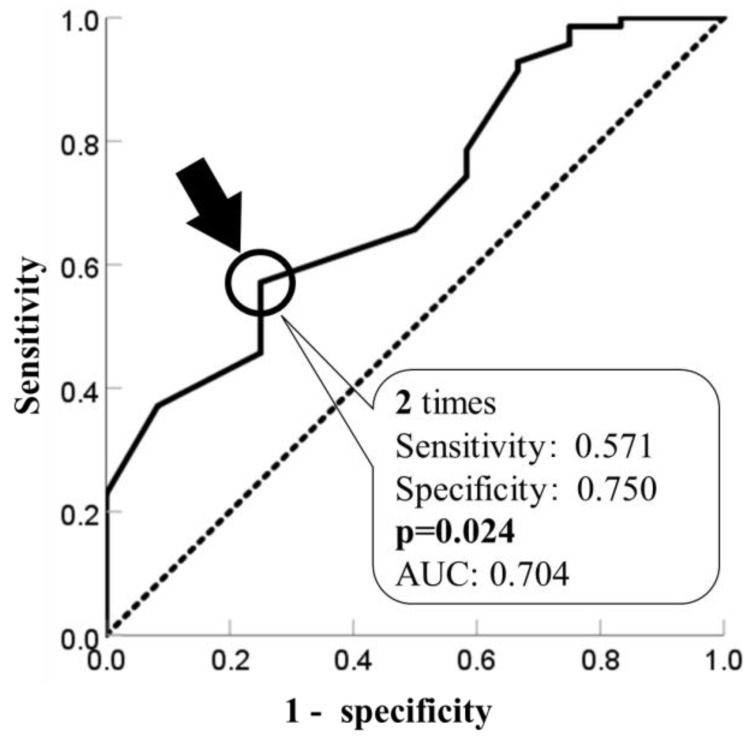
Cut-off value of ΔCS30 for the presence of falls.

**Table 1 healthcare-11-02558-t001:** Characteristics of fall group and no-fall group.

	Fall, *n* = 12			No-Fall, *n* = 70			*Z* or *χ^2^* Value	*p* Value
Fall rate, %	14.6			0				
Age, years	84.8	±	8.1	81.0	±	7.8	1.5 ^b^	0.14
Sex, male, %	41.7			34.3			0.2 ^a^	0.62
Weight, kg	57.3	±	13.3	54.7	±	13.7	0.2 ^b^	0.84
LTCI level, support level 1/2, care level 1/2/3, %	50.0/33.3/16.7/0/0			48.6/41.4/12.9/5.7/1.4			0.9 ^a^	0.91
Comorbidity, %								
Hypertension	41.7			50.0			0.3 ^a^	0.59
Diabetes	16.7			25.7			0.5 ^a^	0.50
Orthopedic disease	41.7			35.7			0.2 ^a^	0.69
Cerebrovascular disease	25.0			30.0			0.1 ^a^	0.73
Parkinson’s disease	16.7			5.7			0.8 ^a^	0.18
Heart disease	8.3			24.3			1.5 ^a^	0.21
Cancer disease	8.3			6.1			<0.1 ^a^	0.89

Values are presented as mean ± standard deviation or %, ^a^: *χ*^2^ value, ^b^: *Z* value. Abbreviations: LTCI, long-term care insurance.

**Table 2 healthcare-11-02558-t002:** Two-group comparison of physical function: at the start, at 6 months, and changes during that period.

	Fall, *n* = 12			No-Fall, *n* = 70			*t* or *Z* Value	*p* Value
Physical function								
Gait speed at start, m/s	0.99	±	0.32	0.93	±	0.33	0.6	0.57
Gait speed at 6 months, m/s	1.34	±	0.41	1.14	±	0.34	1.8	0.07
ΔGait speed, m/s	0.35	±	0.51	0.21	±	0.36	1.2	0.25
TUG at start, s	10.9	±	4.8	13.3	±	8.1	−1.1 ^b^	0.28
TUG at 6 months, s	9.6	±	3.7	11.4	±	6.1	−1.4 ^b^	0.17
ΔTUG, m/s	−1.3	±	2.5	−2.0	±	6.9	−0.3 ^b^	0.77
CS30 at start, times	17.2	±	6.6	12.8	±	7.4	2.0	0.053
CS30 at 6 months, times	15.7	±	6.2	15.3	±	6.1	0.2	0.84
ΔCS30, times	−1.6	±	4.9	2.5	±	4.8	−2.3 ^b^	0.02
Physical function after adjustment								
Gait speed at start, m/s	0.99	±	0.10	0.93	±	0.04	0.6	0.57
Gait speed at 6 months, m/s	1.35	±	0.10	1.14	±	0.05	1.9	0.06
ΔGait speed, m/s	0.36	±	0.1	0.21	±	0.05	1.2	0.10
TUG at start, s	11.3	±	2.3	13.2	±	0.9	−0.8	0.43
TUG at 6 months, s	9.7	±	1.7	11.3	±	0.7	−0.9	0.39
ΔTUG, m/s	−1.6	±	1.9	−1.9	±	0.8	0.2	0.88
CS30 at start, times	16.7	±	2.1	12.8	±	0.9	0.7	0.10
CS30 at 6 months, times	15.7	±	1.8	15.2	±	0.7	0.2	0.81
ΔCS30, times	−1.0	±	1.4	2.4	±	0.5	−2.3	0.03

Values are presented as mean ± standard deviation or %, ^b^: *Z* value. Abbreviations: CS30, 30 s chair stand test; TUG, timed up-and-go test. Analysis of covariance adjustment: age, sex, long-term care insurance.

## Data Availability

Not applicable.

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
