# Peer review of "Differences in Improvement of Physical Function in Older Adults with Long-Term Care Insurance with and without Falls: A Retrospective Cohort Study"

_healthcare, 2023, doi:10.3390/healthcare11182558_

Round 1
Reviewer 1 Report
The authors aimed to examine the relationship between the amount of change in physical function due to exercise and falls during the following six months in older adults with long-term care insurance (LTCI).
In this retrospective longitudinal study, 82 of 96 consecutive daycare center users met the criteria. The participants were divided into two groups based on the presence or absence of falls 6-12 months after use. Participant characteristics in the fall and non-fall groups and physical function at baseline and six months in each group were compared.
The fall group (n = 12, fall rate: 14.6%) had a significantly lower Δ30-s chair stand test (CS30) score than the no-fall group (n = 70) (p = 0.02). Physical functions such as gait speed, timed up-and-go test, and CS30 significantly improved over six months in the no-fall group and in all participants (p < 0.01). Gait speed in the fall group improved significantly over six months (p = 0.04). ΔCS30 was extracted as a predictor of falls (p = 0.022), and its cut-off value was calculated twice (p = 0.024).
The authors concluded that:- the physical function of older adults with LTCI improved after six months of use at a daycare center. -In addition, failure to improvement in CS30 ≥ 2 times after six months of exercise was associated with future falls.
This is a revised manuscript.
I checked the revisions in the body of the manuscript. I found the ms strongly improved.
I have some minor comments:
1. Aim “Therefore, this study aimed to retrospectively examine the differences in…” must be more effective.
2. It is good to arrange the results into themes. Please insert a few rows before presenting them.
3. I suggest to expand a bit the conclusions.
Author Response
レビュアーへの返信 1コメント
これは改訂された原稿です。
原稿本文の改訂を確認しました。msが大幅に改善されたことがわかりました。
私はいくつかのマイナーなコメントがあります:
ポイント1:目的 「そこで、本研究は、日本の違いを後方視的に検討することを目的としていた」より効果的でなければなりません。
応答 1:
コメントありがとうございます。
修正を行いました。
p2、56行目。
そこで本研究の目的は,LTCI高齢者の転倒ありとなしにおける身体機能の違いと変化について,保育園を利用した後の身体機能の違いと変化を後方視的に検討することである。
ポイント2:結果をテーマに並べると良い提示する前に数行を挿入してください。
応答 2:
コメントありがとうございます。
修正を行いました。
p4、155行目およびp5、159行目。
3.2. 転倒群と非転倒群の特徴
表1は、転倒群と転倒なし群の年齢、性別、体重、LTCI、併存疾患などの参加者の特徴を示しています。両群間に有意差はなかった。
3.3. 各グループの身体機能の 事前比較と事後比較
各グループの身体機能の比較前後の結果を以下に示します。ベースライン時と0カ月時の身体機能を比較すると、歩行速度に有意な改善が見られた(93.0±33.1 m/秒 vs. 14.0±34.0 m/sec; p < 001.0, 94.0±32.1 m/sec vs. 17.0±36.0 m/sec; p < 001.13), TUG (3.8±07.11 秒 vs. 4.6±1.0 秒; p = 001.13, 1.7±8.11 秒 vs. 1.5±8.0 秒; p < 001.30), CS12(8.7±4.15倍対3.6±1.0倍;p < 001.13、3.7±5.15倍対3.6±1.0倍、p = 001.0)は、それぞれ転倒なし群およびすべての参加者で。転倒群では,保育所でのプログラム開始から99カ月後に歩行速度に有意な改善が認められた(0.32±1.34 m/sec vs 0.41±0.04 m/sec; p = 30.10)。しかし、TUGおよびCS9では有意な改善は見られなかった(4.8±9.6秒 vs. 3.7±0.06秒;p = 17.2、6 .6±15.7倍 vs. 6.2±0.29倍;p = <>.<>)。
ポイント3:結論を少し広げることをお勧めします。
応答 3:
コメントありがとうございます。
追加しました。
p7、237行目。
デイケアベースの運動プログラム後にΔCS30を<>以上改善しないと、将来の転倒につながる可能性があります。

Reviewer 2 Report
This manuscript entitled "Differences in Improvement of Physical Function in Older Adults with Long-Term Care Insurance With and Without Falls: a Retrospective Cohort Study" examined the differences in physical function and its changes in LTCI older adults with and without falls after using daycare. The report suggests that in older adults with LTCI, physical function with and without falls after daycare use differed by ΔCS30, with a cutoff value of 2.
Methods are presented adequately, and results support the conclusion.
However, there are some minor issues in the manuscript that need to be improved.
Author Response
Response to Reviewer 2 Comments
This manuscript entitled "Differences in Improvement of Physical Function in Older Adults with Long-Term Care Insurance With and Without Falls: a Retrospective Cohort Study" examined the differences in physical function and its changes in LTCI older adults with and without falls after using daycare. The report suggests that in older adults with LTCI, physical function with and without falls after daycare use differed by ΔCS30, with a cutoff value of 2.
Point 1:Methods are presented adequately, and results support the conclusion.
Response 1:
Thank you for your valuable comments.

Reviewer 3 Report
Report can be improved by contextualising what happens at day care centres in Japan. Important detail lacking in methodology and results. E.g., were exercises in groups or customised for individuals? What was the adherence and how was this tracked. Range of dose should be shown (mentioned but vague). Reference changes to MCID would be helpful. The key findings of the results get rather lost in both the results section and the discussion. The key take-home message is not clear.

The English is reasonable, but detail is lacking which makes it difficult to judge the merit of the study.
Author Response
Response to Reviewer 3 Comments
Comments and Suggestions for Authors
Point 1:Report can be improved by contextualising what happens at day care centres in Japan.
Response 1:
Thank you for your comment.
We have corrected the context.
p2, line 49.
However, the fall rate among older adults with LTCI after using daycare was 18.3% over a 3-month period, which is problematic [23]. Additionally, research on the relation-ship between improvements in their physical function and falls is scarce.
- Shimada, H.; Tiedemann, A.; Lord, SR.; Suzukawa, M.; Makizako, H.; Kobayashi, K.; Suzuki, T. Physical factors underlying the association between lower walking performance and falls in older people: a structural equation model. Arch Gerontol Geriatr. 2011, 53(2), 131-134. DOI: 10.1016/j.archger.2010.11.003.
Point 2:Important detail lacking in methodology and results. E.g., were exercises in groups or customised for individuals?
Response 2:
Thank you for your comments. These exercises were performed in groups.
We have appended the content.
p3, line 110.
These exercises were performed in groups.
Point 3:What was the adherence and how was this tracked. Range of dose should be shown (mentioned but vague).
Response 3:
Thank you for your comment.
The purpose of the program and participation rules were explained at the beginning of use, and the staff recorded attendance for each use.
Because we excluded patients with MMSE scores of 24 points or less, it is considered that the adherence group is relatively high. However, we were unable to investigate adherence or its rate.
The six-month participation rate among those analyzed was 83%.
We have added this to the limitation.
We have modified our content.
P3, line 118.
The exercise program was implemented for a six-month period. Additionally, the program’s purpose and participation rules were explained at the beginning of the program. Staff members recorded their attendance for each exercise session.
P4, line 150.
The participation rate in the six-month exercise program was 83%.
P7, line 231.
The compliance rate for the exercise program was not investigated due to missing data; this should be considered while interpreting the results of this study.
Point 4:Reference changes to MCID would be helpful.
Response 4:
Thank you for your comment.
We have appended the MCID to our discussion.
P7, line 228.
Additionally, the minimal clinically important difference (MCID) for CS30 among older adults with orthopedic disease was two times, which is similar to the cut-off value of two obtained in the present study [43].
- Wright, AA.; Cook, CE.; Baxter, GD.; Dockerty, JD.; Abbott, JH. A comparison of 3 methodological approaches to defining major clinically important improvement of 4 performance measures in patients with hip osteoarthritis. J Orthop Sports Phys Ther. 2011. 41(5): 319-27. DOI: 10.2519/jospt.2011.3515.
Point 5:The key findings of the results get rather lost in both the results section and the discussion. The key take-home message is not clear.
Response 5:
Thanks for your comment.
We added content.
P5, line 172.
Table 2 presents a comparison of physical function between the fall and no-fall groups.
P7, line 233.
In clinical practice, an increase in CS30 by two or more times from the start of an exercise program may be important for preventing falls among older adults.
Comments on the Quality of English Language
The English is reasonable, but detail is lacking which makes it difficult to judge the merit of the study.
Thank you for your comment on the file.
We have modified the content.
Point 6
not well phrased and if one is not familiar with day care use difficult to contextualise
Response 6:
Thanks for your comment.
Fixed some.
p1, line 14.
This study examined the differences in changes in physical function with and without falls after daycare use among frail older adults with long-term care insurance (LTCI).
Regarding the explanation of LTCI, it is difficult to explain with an abstract of limited words, so I explained it in 「2.2. LTCI levels」 of methods in the main text.
Point 7
Receiver?
Response 7:
Thanks for your comment.
Fixed.
Point 8
such as?
Response 8:
Thanks for your comment.
Fixed.
Moreover, it presents serious problems such as increase in healthcare and long-term care expenditures that are associated with hospitalization and LTCI use [6].
Point 9
very difficult to know what daycare comprises?
Response 9:
Thanks for your comment.
The primary target audience for day care is older adults from Support level 1 to Care level 1.
The main content is to maintain and improve mobility, balance and muscle strength. Those contents were described in [2.5. Exercise program] of Methods.
Point 10
Was participant given a meter before timing started so preferred walking speed was in fact measured? And continued walking for 1m after timing stopped?
Response 10:
Thanks for your comment.
Normal gait speed was measured by a physiotherapist using a stopwatch. A 1m acceleration and deceleration walkway was set before and after the 5m walkway.
we made a fix.
[2.4. Physical function]
Gait speed, a measure of walking ability, was assessed by a physical therapist using a stopwatch and was defined as the time required to walk 5 m at a normal speed [27]. A 1 m acceleration and deceleration walkway was set up before and after the 5 m walkway.
Point 11
very vague. Were these group or individual sessions?
Response 11:
Thanks for your comment.
It will be a group session.
we made a fix.
[2.5. Exercise program]
These exercises were performed in groups.
Point 12
Vague
Response 12:
Thank you for your comment.
We have made some corrections.
[2.5 Exercise program]
Concerning exercise intensity, the rating of perceived exertion (RPE) was “fairly light to somewhat hard” for aerobic exercise [31], and “approximately 60% intensity of maximal load × 10–15 times × three sets” for resistance training [32].
Point 13
More detailed required here - for example, consider presence of Parkinson's disease which would impact balance and gait
Response 13:
Thanks for your comment.
Parkinson's disease occurred in 16.7% of the fall group and 5.7% of the no-fall group (significant difference p=0.18).
We have revised the table.

Reviewer 4 Report
What was the study design? Why it was a retrospective cohort study. You recruited 82 participants which falls and without falls by history of falling withing 6-12 months and then divided into falls and without falls. After that, all participants were received exercise program and then physical function were examined (both before and after intervention). is that right? It was confused. Please clarify.
Exercise program; line 100-110. Please provide the duration of the exercise program.
Sample size; line 112-114. You did the sample size calculation and it was at least 36 that is a sufficient number of the participants for the study. Therefore, why did you recruit 82 participants? Additionally, the sample size should requires equal sample size in each group (i.e., falls and non-falls).
Discussion part. Should provide the reason why a significant improvement in CS30, but not in gait speed and TUG.
Fair.
Author Response
Response to Reviewer 4 Comments
Comments and Suggestions for Authors
Point 1:What was the study design?
Response 1:
Thanks for your comment.
The study design was a retrospective cohort study.
Point 2:Why it was a retrospective cohort study. You recruited 82 participants which falls and without falls by history of falling withing 6-12 months and then divided into falls and without falls.
After that, all participants were received exercise program and then physical function were examined (both before and after intervention). is that right? It was confused. Please clarify.
Response 2:
This study design was not an RCT.
We considered the presence or absence of falls after an exercise program to be important.
At a daycare facility, we investigated the presence of falls after day care use in LTCI older people undergoing an exercise program.
All of the daycare users included in the study had been in the exercise program for at least six months.
The difference between the presence or absence of subsequent falls and the change in physical function over the six-month period was unknown.
Therefore, the purpose of this study was to investigate differences in the presence or absence of falls and changes in physical function in all populations that underwent an exercise program.
We have appended the content.
P2, line 51.
Furthermore, although it is essential to track the presence or absence of falls after an exercise program, few studies have investigated the differences in changes in physical function between the presence and absence of falls.
Point 3:Exercise program; line 100-110. Please provide the duration of the exercise program.
Response 3:
Thanks for your comment.
We added the period.
P3, line 118.
The exercise program was implemented for a six-month period.
Point 4:Sample size; line 112-114. You did the sample size calculation and it was at least 36 that is a sufficient number of the participants for the study. Therefore, why did you recruit 82 participants? Additionally, the sample size should requires equal sample size in each group (i.e., falls and non-falls).
Response 4:
Thanks for your comment.
The study design was a retrospective cohort study, and serial users had to be enrolled. Fall are likely to be fewer than no-fall, so we employed more users than the sample size.
Point 5:Discussion part. Should provide the reason why a significant improvement in CS30, but not in gait speed and TUG.
Response 5:
Thanks for your valuable comments.
In table2 of the results, Δgait speed and ΔTUG showed no difference in change in physical function between the fall and no-fall groups.
It was inferred that the six-month exercise program had a similar effect on gait speed and TUG in both groups.
The following information has been added to the text.
P6, line 217.
In this study, changes in gait speed and TUG did not differ between the fall and no-fall groups. In a previous randomized controlled trial (RCT) among older adults with LTCI, an exercise program increased walking ability and balance [38]. Therefore, it is possible that the present study had similar effects on gait speed and TUG in both groups.
- Shinohara, H.; Mikami, Y.; Kuroda, R.; Asaeda, M.; Kawasaki, T.; Kouda, K.; Nishimura, Y.; Ohkawa, H.; Uenishi, H.; Shimokawa, T.; Mikami, Y.; Tajima, F.; Kubo, T. Rehabilitation in the long-term care insurance domain: a scoping review. Health Econ Rev. 2022. 12(1): 59. DOI: 10.1186/s13561-022-00407-6.

Round 2
Reviewer 3 Report
I appreciate the responses to some of the concerns, but still feel there are issues with the overall paper linked to the design.
Improved.
Author Response
We have responded to the attached file.
